# Parallelism Strategies for Big Data Delayed Transfer Entropy Evaluation

**Jonas R. Dourado, Jordão Natal de Oliveira Júnior and Carlos D. Maciel \***

Department of Electrical and Computational Engineering, University of São Paulo, 13566-590 São Carlos-SP, Brazil; jonas.jonaias@gmail.com (J.R.D.); jordao.oliveira@usp.br (J.N.d.O.J.)

\* Correspondence: carlos.maciel@usp.br

**Abstract:** Generated and collected data have been rising with the popularization of technologies such as Internet of Things, social media, and smartphone, leading big data term creation. One class of big data hidden information is causality. Among the tools to infer causal relationships, there is Delay Transfer Entropy (DTE); however, it has a high demanding processing power. Many approaches were proposed to overcome DTE performance issues such as GPU and FPGA implementations. Our study compared different parallel strategies to calculate DTE from big data series using a heterogeneous Beowulf cluster. Task Parallelism was significantly faster in comparison to Data Parallelism. With big data trend in sight, these results may enable bigger datasets analysis or better statistical evidence.

**Keywords:** delayed transfer entropy; parallelism strategies; big data analysis; heterogeneous computer cluster; complex systems; causality; surrogate

## 1. Introduction

Recently, the amount of data being generated and collected have been rising with the popularization of technologies such as Internet of Things, social media, and smartphone [1]. The increasing amount of data led the creation of the term big data, with one definition given by Hashem et al. [2], as a set of technologies and techniques to discover hidden information from diverse, complex and massive scale datasets. One class of hidden information is causality, which Cheng et al. [3] discuss and propose a framework to deal with commonly found big data biases such as confounding and sampling selection.

The interaction among random variables and subsystems in complex dynamical multivariate models as seen in big data are a developing research area that had been widely applied in a variety of fields, such as climatic processes [4,5], brain analysis [6,7], pathology propagation [8,9], circadian networks [10,11], among others. The knowledge about causality in such complex systems can be useful for the evaluation of their dynamics and also to their modelling since it can lead to topological simplifications. In many cases, the assessment of interaction or coupling is subject to bias [12].

Among the tools to infer causal relationships, there is Mutual Information used by Endo et al. [13] to infer neuron connectivity by [14] to optimization of meteorological network design. Additionally, Transfer Entropy (TE) exists, which allows identification of a cause–effect relationship by not accounting for straightforward and uniquely shared information [15]. TE has been applied to many complex problems from diverse research fields e.g., oscillation analysis [16], finance [17,18], sensors [19–22], biosignals [23,24], thermonuclear fusion [25], complex networks [26], geophysical phenomena [27,28], industrial energy consumption network [29] and algorithmic theory of information [30]. In addition, TE has been implemented in non-Gaussian distributions, such as: multivariate exponential, logistic, Pareto (type I–IV) and Burr distributions [31].

A derivation of TE metric called Delayed TE (DTE) is explored in Kirst et al. [32] to extract topology in complex networks, where they mentioned an ad hoc complex network example. Liu et al. [33]

make inferences in nonlinear systems with this tool; in addition, Berger et al. [34] use DTE to estimate externally applied forces to a robot using low-cost sensors. However, DTE calculation requires a high demanding processing power [35,36], which is aggravated with large datasets as those found in big data. Many approaches were proposed to overcome such performance issues, e.g., an implementation using a GPU made by Wollstadt et al. [37] and an implementation using an FPGA made by Shao et al. [36].

Parallel programs should be optimised to extract maximum performance from hardware on architecture case by case [38], which is far from trivial according to Booth et al. [39]. There exist different and combined manners to explore parallelism such as Data Parallelism and Task Parallelism [40]. Choudhury et al. [41] stated that choosing the configuration of parallel programs is a "mysterious art" in a study in which they created a model aiming at maximum speedup by balancing different parallelism strategies for both cluster and cloud computing environments. Moreover, Yao and Ge [42] discuss the importance of parallel computation using big data in industrial sectors.

In this paper, we address DTE performance issue by using a previously not described approach to decrease DTE execution time using a Beowulf cluster; then, we present and execute two parallelism strategies on big data time series and compare run time difference. An additional importance support to this theme is given by Li et al. [35], where they concluded that TE performance is about 3000 times slower than Granger Causality, making TE unsuited for many Big Data analysis. Finally, we analyze computing node performance during Task Parallelism to gain some insights to enrich parallel strategies discussion.

This paper is organized as follows: the initial concepts section presents topics that might help readers keep up with the whole paper. In Materials and Methods, all steps needed to reproduce the results are presented. Results and Discussion show the performance of algorithms and cluster nodes during a large neurophysiology database analysis. In conclusion, additionally, future works are suggested. Remaining information useful for reproducibility is located in an appendix to avoid nonessential noise through the text reading.

## 2. Initial Concepts

The term Big Data has a variety of definitions [2,43–46], one proposed by the National Institute of Standards and Technology stating where data volume, acquisition rate or representation demand nontraditional approaches to data analysis or requires horizontal scaling for data processing [45]. Another definition is proposed by Hashem et al. [2] as a set of technologies and techniques to discover hidden information from diverse, complex and massive scale datasets.

The most important phase of Big Data according to Hu et al. [45] is in the analysis, where it has the goal of extracting meaningful and often hidden [2] information. Analysis phase importance is reinforced by Song et al. [47], which affirms that one of the Big Data value perspectives lies in analysis algorithms.

### 2.1. Computer Cluster

Software long run time is often an issue, especially in big data context [48]. To decrease software experiment execution time, one can use faster hardware or optimize underlining algorithms. Some hardware options to decrease execution time include FPGA [49,50], GPU [51], faster processors [52] or computer clusters [53,54]. Algorithm optimization examples are found in studies by Gou et al. [55], Naderi et al. [56] and Sánchez-Oro et al. [57].

Among computer clusters, one well-known implementation is Beowulf cluster, made by connecting consumer grade computers on a local network using Ethernet or another suitable connection technology [58]. The term Beowulf cluster was coined by Sterling et al. [59], which created the topology on NASA facilities as a low-cost alternative to an expensive commercial vendor built form High-Performance Clusters. Beowulf cluster is widely used by diverse research fields such as Monte Carlo simulations [60], drug design [61], big data analysis [48] and neural networks [62].

According to Booth et al. [39], archiving parallel performance on chosen hardware architecture depends on factors such as scheduler overhead, data/task granularity, cache fitting and data synchronization. There exist different abstraction levels of parallelism strategies that can be combined [41]. Often, a systematic comparison between parallelism strategies is necessary to verify which one has better performance [39]. A comparison of two parallelism strategies in a Beowulf cluster is shown by [63].

Data Parallelism strategy, as stated by Gordon et al. [40], is when one processing data slice does not have dependency on the next one. Thus, data are divided into several data slices and processing them equally by different processors. The Task Parallelism objective is to spawn tasks across processors to speedup one scalable algorithm. Tasks can be spawned by a central task system or by a distributed task system, both adding processing overhead, with a distributed task system achieving less overhead [39].

### 2.2. IPython Parallel Environment

IPython was born as an interactive system for scientific computing using the Python programming language [64], later receiving several improvements as parallel processing capabilities [65]. Recently, these parallel processing capabilities become an independent package under IPython project and were renamed as ipyparallel [66].

Ipyparallel enables a Python processing script to be distributed across a cluster, with minor code modifications [67]. Throughout the text, ipyparallel is referenced as IPython, since its documentation also refers to itself as IPython. IPython already had been used in studies similar to our purpose, Kershaw et al. [68] used it for big data analysis in a cloud environment and Stevens et al. [69] developed an automated and reproducible neuron simulation analysis.

### 2.3. Algorithms

At this subsection, the algorithms used during the parallel DTE evaluation are presented. The first one is *surrogate* algorithm that was used to infer significance level in our analysis. The second was *delayed transfer entropy* used to estimate the causal interaction among signals presented in our dataset.

#### Surrogate

The word *surrogate* stands for something that is used instead of something else. In the case of surrogate signals [70], the synthetic data used are randomly generated, but it also presents some characteristics of the original signal that is taking place. A surrogate has the same power spectrum that the original data, but these two signals are uncorrelated. Different computational packages present algorithms to generate surrogate signals [71,72]. The Amplitude Adjusted Fourier Transform (AAFT) method explained in Lucio et al. [73] proposes rescaling the original data to a Gaussian distribution using fast Fourier transform (FFT) phases' randomization and inverse rescaling. This procedure introduces some bias and Lucio et al. [73] showed a method to remove it by adjusting the spectrum from surrogates, named Iterative Amplitude Adjusted Fourier Transform (IAAFT) Lucio et al. [73] and is displayed in Algorithm 1.

Surrogate data represent, as best as possible, all the characteristics of the real process, though without causal interactions.

---

**Algorithm 1** IAAFT

---

  1:  original_frequency_bins ← FFT(original_signal)
  2:  original_frequency_amplitute ← abs(original_frequency_bins)
  3:  original_frequency_phase ← phase(original_frequency_bins)
  4:  permuted_signal ← random_permutation(original_signal)
  5:  permuted_frequency_bins ← FFT(permuted_signal)
  6:  permuted_frequency_amplitude ← abs(permuted_frequency_bins)
  7:  **while** num_step < max_steps **do**

  8:      permuted_frequency_phase ← phase(permuted_frequency_bins)
  9:      new_frequency_bins ← original_frequency_amplitude*exp(j*permuted_frequency_phase)
10:     new_signal ← IFFT(new_frequency_bins)
11:     permuted_signal ← swap_vaues(original_singal, new_signal)
12:     permuted_frequency_bins ← FFT(permuted_signal)
13:     permuted_frequency_amplitude ← abs(permuted_frequency_bins)
14:     current_error ← error(permuted_frequency_amplitude,original_frequency_amplitude))
15:     **if** num_step = 0 **then**

16:        min_error = current_error
17:     **end if**
18:     **if** current_error < min_error **then**

19:        min_error = current_error
20:        best_result = real(new_signal)
21:     **end if**
22:     **if** num_step > 0 **and** abs(last_error-current_error) < stop_tolerance **then**

23:        return best_result
24:     **end if**
25:  **end while**

---

In the case of neurophysiological data, the causal association happens in phase synchronization [74]. Endo et al. [13] used a surrogate data with the IAAFT (Iterative Amplitude Adjusted Fourier Transform) algorithm [75], which generates signals preserving the power density spectrum and probability density functions, but with the phase components randomly shuffled [76].

The importance of surrogate repetitions number is explicitly derived from Equation (1) [77],

$$n = \frac{K}{\alpha} - 1, \tag{1}$$

where the amount of surrogate repetitions $n$ is inverse proportional to the desired significance level $\alpha$, and $k = 1$ for a one-sided hypothesis test or $k = 2$ for a two-sided hypothesis test. A full diagram of surrogate with DTE is presented in Figure 1.

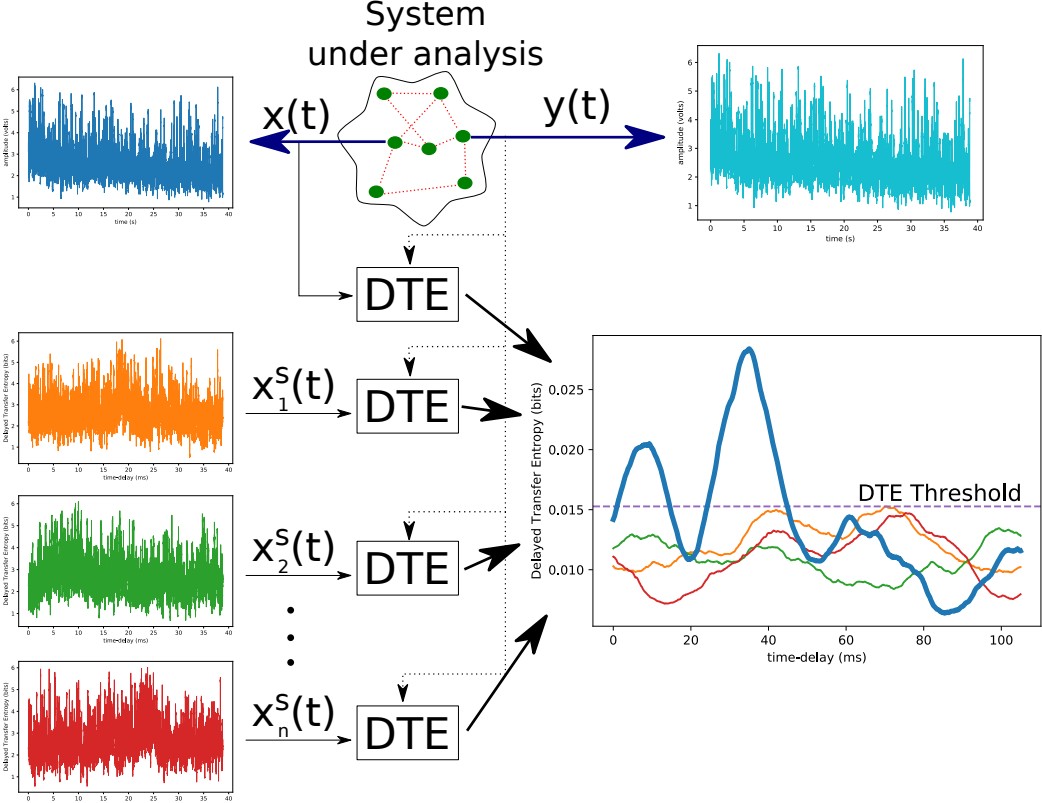

**Figure 1.** Data analysis and significance levels. From input data, a series of surrogate signals $X_n^S(t)$ is estimated, and for each of the surrogate signals, time delayed transfer entropy is determined. The **n** repetitions indicate a significance level from Equation (1) against random probabilities with the same power spectra and amplitude distribution as real data. Assuming the number of surrogate **n** equals three as shown in this figure, DTE Threshold line displayed on the biggest ensemble plot has a significance level of 75%.

## 3. Material and Methods

This study emerged from recurrent cluster usage in our laboratory, demanded by several applications as multi-scenarios' Monte Carlo simulations [78], optimization of large-scale systems reconfiguration [79], and in specific biosignals analysis with DTE [80]. Data used in those studies were made from intracellular multi-recording and large scale power distribution system. In both cases, the database was composed of tens of thousands of signals (up to two million data samples each) or thousands of nodes in a power distribution system.

*Transfer Entropy*

Transfer Entropy (TE) measurement, shown in Equation (2), was introduced by Schreiber [15] and is used to measure information transfer between two time series. TE has an asymmetric nature, it being possible to determine information direction [81], is

$$TE_{X \rightarrow Y} = \sum_{y_{n+1}, y_n, x_n} p(y_{n+1}, y_n^{(k)}, x_n^{(l)}) \log_2 \frac{p(y_{n+1}|y_n^{(k)}, x_n^{(l)})}{p(y_{n+1}|y_n^{(k)})}, \qquad (2)$$

where $y_n$ and $x_n$ denote values of $X$ and $Y$ at time $n$; $y_{n+1}$ the value of $Y$ at time $n + 1$; $p$ is the probability of parenthesis content; $l$ and $k$ are the number of time slices used to calculate probability density function (PDF) using past values of $X$ and $Y$, respectively; chosen $log_2$ means that TE results are given in bits.

Assuming $k = 1$ and $l = 1$ to simplify analysis (also called as D1TE by Ito et al. [82]), the TE algorithm is demanding regarding computational power [36], with its computational complexity being $O(B^3)$ [36], where $B$ is the chosen number of bins in PDF.

An extension to D1TE proposed by Ito et al. [82] is delayed transfer entropy (DTE—Equation (3)), which is a D1TE with variable causal delay range. This way, a parameter $d$ represents a variable delay between $y$ and $x$. DTE is a useful metric to determine where, within $d$ range, the biggest transfer of information from $X$ to $Y$ occurs. DTE is defined as

$$DTE_{X \to Y}(d) = \sum_{y_{n+1}, y_n, x_{n+1-d}} p(y_{n+1}, y_n, x_{n+1-d}) \log_2 \frac{p(y_{n+1}|y_n, x_{n+1-d})}{p(y_{n+1}|y_n)}, \tag{3}$$

where $y_n$ denotes value of $Y$ at time $n$; $y_{n+1}$ the value of $Y$ at time $n+1$; $x_{n+1-d}$ denotes value of $X$ at time $n+1$ delayed by parameter $d$; $p$ is the probability of parenthesis content; chosen $log_2$ means that DTE results are given in bits.

The fundamental reason for which DTE was used instead of Mutual Information (MI) is that, for the biosignals data and the power analysis mentioned above, it was important to know the direction of the information flow, and this property is achieved by TE (and DTE) but not by MI, since $MI(X; Y) = MI(I; X)$ [83].

Studies with DTE usage were negatively affected by high processing power demands [36], therefore often limiting data size scope or even number of surrogate datasets for DTE analysis. With an objective to clarify program flow, the serial version of DTE analysis is shown in Algorithm 2. The program calculates embedding parameters for each channel and calculates DTE for each two channel permutation. Finally, surrogate signals representing each permutation are generated, and DTEs are calculated. The more surrogates, the better, since it contributes to increasing causality statistical evidence.

---

**Algorithm 2** Execute DTE with surrogate

---

1: **for** experiment in experiment_list_file **do**
2:     experiment_signal ← load_signal_from_disk (experiment)
3:     **for** each channel in experiment_signal **do**
4:         calculate embedding(channel)
5:     **end for**
6:     **for** each two-channel permutation in experiment_signal **do**
7:         calculate DTE(channel1, channel2, channel2_embedding)
8:     **end for**
9:     **for** $i = 0$ to num_surrogates **do**
10:         surrogate ← generate_surrogate (experiment_signal)
11:         **for** each two-channel permutation in surrogate **do**
12:             calculate DTE(channel1, channel2, channel2_embedding)
13:         **end for**
14:     **end for**
15: **end for**

---

Our surrogate algorithm (Algorithm 1 uses FFTW library [84] to archive the best performance during FFT and inverse fast Fourier transform (IFFT) routines using arbitrary-size vector transforms. Embedding was calculated to find the target variable's past, which can be found in the first local minimum of auto-mutual information or first zero crossing auto-correlation measures [85].

The previous existent code comes from several years of development, is arbitrarily named as CODE A and is shown as a flowchart in Figure 2a. CODE A is a Python [86] script which uses libraries such as numpy [87], IPython [64] and lpslib (developed in-house by LPS laboratory). It tackled performance issues by using Data Parallelism, made possible by an IPython *ipyparallel* library.

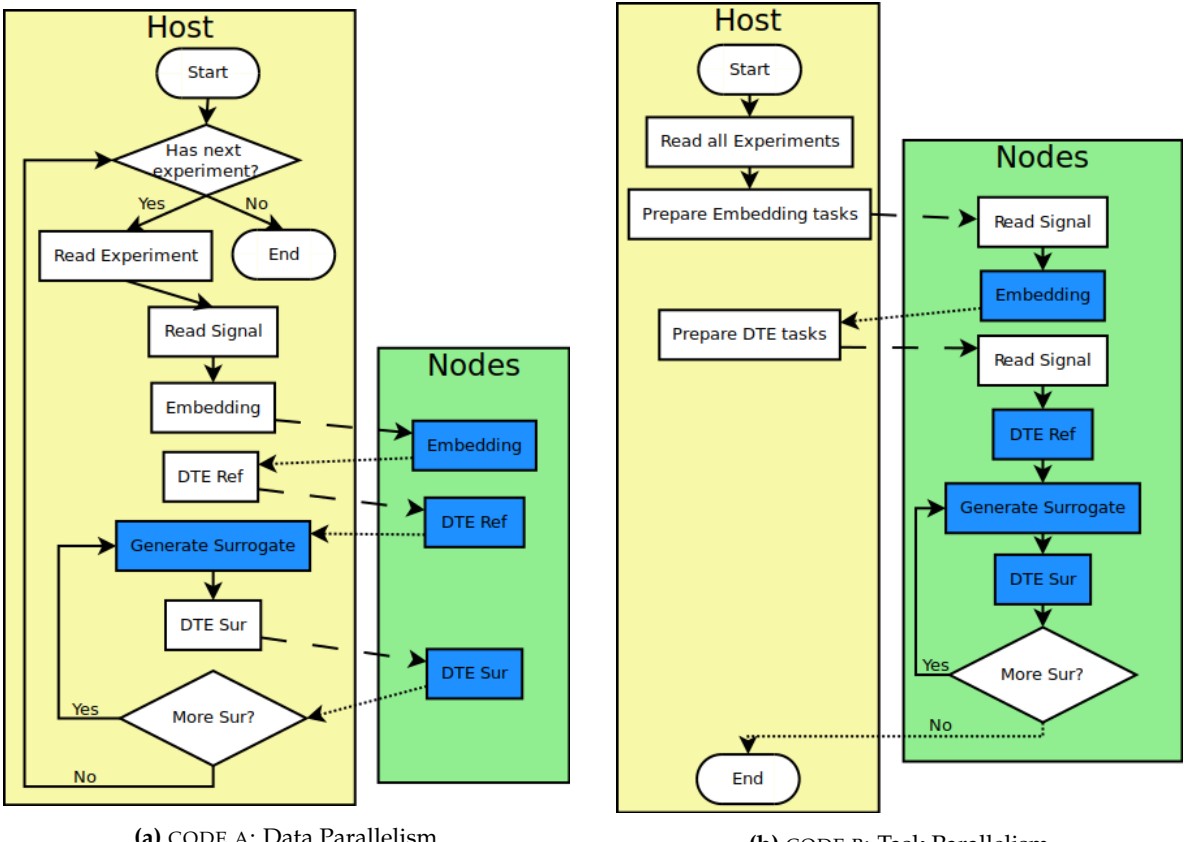

**(a)** CODE A: Data Parallelism                              **(b)** CODE B: Task Parallelism

**Figure 2.** Data and Task Parallelism algorithm flowcharts. Note that each dashed line means data transfer to every cluster node and every dotted line means data gathering and synchronization to host node. Objects of this study are highlighted in blue.

On CODE A, for each experiment, signals are read on the host node, the embedding code is executed on the host node, embedding pushes corresponding channel data to all computing nodes, and each computing node processes one part of the data and results are gathered on the host node. Note that embedding is executed once for each channel. After embedding, DTE is executed on the host node for each two channel permutation; it pushes channel data to all computing nodes, each node processes one part of the data and results are gathered on the host node.

The decision to try another parallelism strategy comes from analyzing CPU load during CODE A execution and observing node processors' sub-utilization. This observation was done by executing *htop* [88] program on a computing node and checking that system load average was significantly smaller than the number of processors. By reading Linux *proc* manual [88], DTE calculation was not in the run queue or waiting for disk I/O to fully load (meaning the load average is equal to the number of processors) each node.

Further investigation by adding extensive logging facilities to CODE A confirmed that the low load average was caused by network communication bottleneck. The code was refactored to use task-based parallelism, aiming to mitigate communication bottleneck. The idea is to load signal data from local storage instead of network transfer from host node. Refactored code was named as CODE B and is shown in Figure 2b.

Before executing CODE B, every signal data must be copied to each computing node. The main difference from CODE A is that every experiment is read once, every embedding task is executed, and, finally, one task is created for each two-channel permutation for every experiment. Tasks are asynchronously executed and scheduled by IPython.

Both CODE A and CODE B were executed with a different number of surrogates (1, 5, 10 and 20) to compare performance between them, except 20 surrogates for CODE A due to excessively long execution time (estimated in more than 6000 min by extrapolating results from CODE A with a smaller number of surrogates). To give some data size perspective, for Data and Task Parallelism experiments involving ten surrogates, the total number of calculated TEs is about 11,642,400 (35 signals × 12 channel pairs (2-permutations of four channels) × 2520 TE/permutation (refers to variation of DTE $d$ delay parameter) × (1 original piece of data + 10 surrogate)).

The system used in the experiments to analyze performance difference was a heterogeneous Beowulf cluster (Figure 3) composed of 10 nodes connected through Gigabit Ethernet Switch model HP Procurve 1910-24G. Node hardware configuration is listed in Table A2 and software configuration is listed in Table A3, moreover, are presented in the Appendix A.

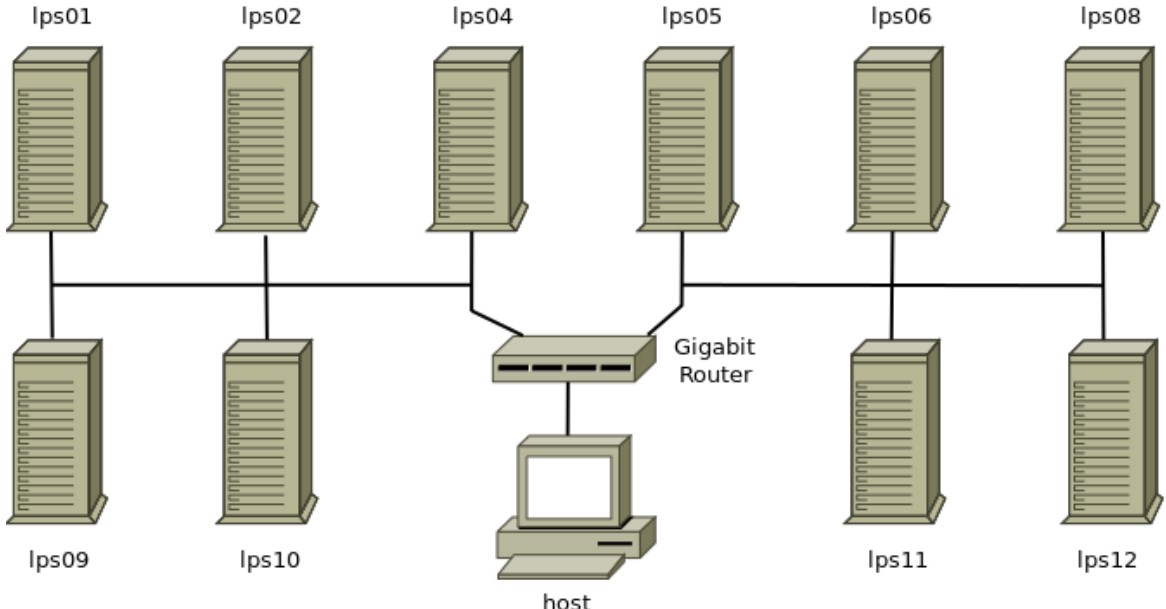

**Figure 3.** Beowulf cluster used in the experiments. Lines between devices may represent multiple Ethernet cables for diagram cleanliness purposes. Cluster nodes are prefixed with "lps". Detailed cluster configuration is informed in the Appendix A.

Logs were processed to calculate duration for each execution. A linear least square method was used to fit a line for data and Task Parallelism duration. By supposing surrogate creation is insignificant in comparison with DTE duration, each line slope represents minutes/surrogate. Finally, the quickening (This word has been chosen since it has the same meaning as the world "speedup", but the later one has a precise definition within Parallel Computing, and the speedup being measured here is not the same as the definition.) was calculated using line slopes to measure performance gain from Task Parallelism over Data Parallelism.

## 4. Results and Discussion

Execution logs gave the total execution time per number of surrogates as shown in Figure 4 by the point marks, and the presented dashed lines are fitted by the linear least square method. Line slopes for each parallelism strategy are 280.385 min/surrogate (Data Parallelism) and 65.257 min/surrogate (Task Parallelism). Therefore, the quickening can be determined as

$$Quickening = \frac{t_{Data\ Parallelism}}{t_{Task\ Parallelism}} = \frac{280.385}{65.257} = 4.297. \tag{4}$$

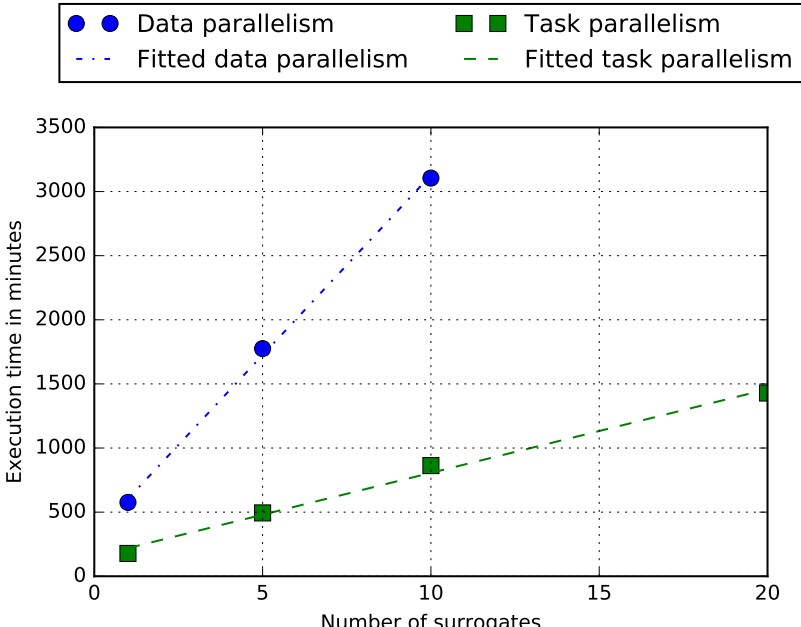

**Figure 4.** Data and Task Parallelism total execution time per number of surrogate signals. Point marks show numerical results in minutes. Lines show data fitted by the linear square method.

The achieved ∼4.3 quickening shows that Task Parallelism is significantly faster than Data Parallelism. After analyzing logs, positive quickening can be explained by three main factors, and it is negatively impacted by another.

First, quickening explanation is data locality since data are stored on a local disk in Task Parallelism versus being transferred by the network in Data Parallelism. In addition, the former has to transfer channel data for every surrogate, while, the latter, locally read signal data only once for each two channel permutation surrogates.

The second factor is sub-optimum node utilization caused by cluster heterogeneity illustrated in Figure 5. This happens in Data Parallelism because data are equally divided across computing nodes with different performance, causing faster nodes, which finished data processing, to wait for slower nodes.

The third factor is caused by the fact of Data Parallelism surrogate datasets are generated only by host node, forcing all computing nodes to wait for surrogate dataset generation. Task Parallelism does not suffer from the same problem, as while one surrogate dataset is generated by one computing nodes, it does not block another computing node.

The asynchronous nature of tasks are negatively affecting Task Parallelism quickening, when task pool is exhausted, some computing nodes are left without any task, reducing the quickening enhancement.

About node performance difference, Figure 5 suggests that it may be caused by different random-access memory (RAM) sizes across cluster nodes, since slowest nodes (*lps02*, *lps04*, *lps05* and *lps08*) have the smallest RAM amount (8 GiB, 8 GiB, 8 GiB and 12 GiB, respectively) as listed in Table A2 in Appendix A.

An analogy can be made between MapReduce and presented parallelism strategies. In Figure 2a,b, dashed and doted arrows would correspond to respectively map and reduce operations. Keeping the same analogy, this study would be about the balance between communication and computation to optimize MapReduce DTE runtime performance.

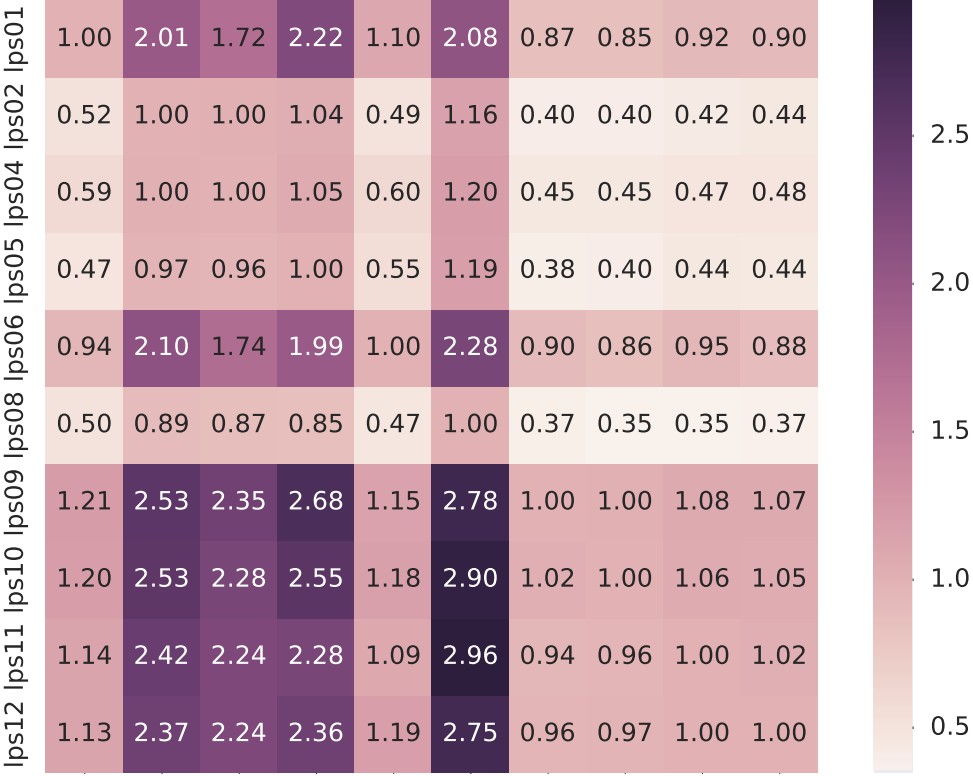

**Figure 5.** DTE Task Parallelism performance comparison between different computing nodes to highlight cluster heterogeneity. In Task Parallelism, each experiment spawned *number of channels* ∗ (*number of channels* − 1) DTE tasks of the same size across computing nodes, their execution times were used to calculate quickening of the *y*-axis computing node over the *x*-axis computing node and finally an average of calculated speedups for each cell was made. An execution log used to generate this plot was from 20 surrogates. Values are given in relative quickening.

## 5. Conclusions

DTE is a probabilistic nonlinear measure to infer causality within a time delay between time-series. However, the DTE algorithm demands high processing power, requiring approaches to overcome such limitation. A distributed processing approach was presented to accelerate DTE computation using parallel programming over a heterogeneous low-cost computer cluster. In addition, Data and Task Parallelism strategies were compared to optimize software execution time.

The main contribution of this study is exploring a low-cost Beowulf heterogeneous computer cluster as a new alternative to existent FPGA TE [36] or GPU DTE [37] implementations. The low-cost nature of Beowulf computer clusters and its simple setup enables using existing computers from research laboratories or universities, helping mitigate DTE performance issues without the acquisition of expensive hardware such as FPGAs or GPU cards. This is especially attractive to places where research funding lacks enough resources or where DTE usage is too infrequent to justify any purchases.

Additionally, in order to verify cluster feasibility, using a Task Parallelism strategy to increase DTE algorithm performance in a heterogeneous cluster was shown as a faster alternative in comparison to Data Parallelism. Having in sight big data analysis importance, it is a significant result, since it will enable causal inference for bigger previous inapplicable datasets or with better causality statistical evidence.

Having verified Task Parallelism as a better approach to DTE in a heterogeneous Beowulf cluster, it remains open how the number of computing nodes affects performance. Thus, future research

should investigate how scalable Task Parallelism is after an increased number of computing nodes observing Amdahl's law [89]. Along the same lines, performance, scalability and cost analysis of renting Cloud Computing nodes to build a cluster on demand to explore causality in big data using DTE is needed.

Studying DTE applied to Big Data demands high processing power, for example, to increase confidence from 95.24% (*n* = 20) to 99.9% (*n* = 999), Task Parallelism run time is increased from 0.9 days to estimated 45.27 days using our setup and data. Although a long runtime is a notorious improvement from about half a year from extrapolated Data Parallelism run time. This highlights the importance of hardware performance to increase statistical confidence and gives strong support to keep researching quicker methods for DTE.

One open question unique to our cluster configuration is if the RAM amount has a correlation with performance in computing nodes when executing Task Parallelism as suggested by our results. Moreover, different parallelism strategies can be tested on a case by case basis aiming to accelerate processing of the ever increasing data size.

**Author Contributions:** Conceptualization, J.R.D.; Funding acquisition, J.R.D. and C.D.M.; Investigation, J.R.D. and J.N.d.O.J.; Methodology, Michel Bessani; Resources, C.D.M.; Software, C.D.M.; Supervision, C.D.M.; Writing—original draft, J.R.D. and J.N.d.O.J.; Writing—review & editing, J.N.d.O.J.

**Funding:** This research was partially funded by **CNPq**–Project 465755/2014-3; **FAPESP** Project 2014/50851-0; **BPE Fapesp** 2018/19150-6.

**Acknowledgments:** The authors would like to thank previous people who worked on source code of the previous version.

**Conflicts of Interest:** The authors declare no conflict of interest.

## Appendix A

To address recent concerns about experiment reproducibility across every science field [90,91], this paper aims to achieve complete reproduction. Therefore, in this appendix, additional information useful to reproduce it is shown.

*Appendix A.1. Source Code*

All code was managed using Git version control software within a private repository. Exact code revisions employed by this study are shown in Table A1.

**Table A1.** Code Git revisions hash.

| Parallel Strategy | Revision Hash |
| --- | --- |
| Data Parallelism | f85aac7e8ff46c74b8e758211197dfc8b069571d |
| Task Parallelism | e97a687c51cfad61ac097fb5fc26b029967615da |

*Appendix A.2. Cluster Configuration*

Cluster hardware configuration is listed in Table A2 and software configuration is listed in Table A3.

**Table A2.** Cluster hardware configuration. RAM modules were listed separately since some nodes have multiple memory modules to explore dual channels. Main storage describes storage media used during script execution, and some nodes might have other unused storage media.

| Node | Processor (cores) | RAM (speed) | Main Storage Size (model) | Ethernet |
|------|-------------------|-------------|---------------------------|----------|
| host | i5-2500 CPU @ 3.30GHz | 4 + 4 GiB (1333MHz) | 2TB WDC WD20EARX-00P | Gigabit |
| lps01 | i7-4770 CPU @ 3.40GHz (8) | 8 + 8 GiB (1333MHz) | 1TB ST1000DM003-1CH1 | Gigabit |
| lps02 | i7-3770 CPU @ 3.40GHz (8) | 8 GiB (1333MHz) | 60GB KINGSTON SV300S3 | Gigabit |
| lps04 | i7-4820K CPU @ 3.70GHz (8) | 8 GiB (1333MHz) | 2TB ST2000DM001-1CH1 | Gigabit |
| lps05 | i7-4820K CPU @ 3.70GHz (8) | 8 GiB (1333MHz) | 1863GiB ST2000DM001-1CH1 | Gigabit |
| lps06 | i7-4820K CPU @ 3.70GHz (8) | 8 + 8 GiB (1333MHz) | 60GB KINGSTON SV300S3 | Gigabit |
| lps08 | i7 950 CPU @ 3.07GHz (8) | 4 + 4 + 4 GiB (1066MHz) | 2TB ST32000542AS | Gigabit |
| lps09 | i7-4790 CPU @ 3.60GHz (8) | 8 + 8 GiB (1600MHz) | 256GB SMART SSD SZ9STE | Gigabit |
| lps10 | i7-4790 CPU @ 3.60GHz (8) | 8 + 8 GiB (1600MHz) | 256GB SMART SSD SZ9STE | Gigabit |
| lps11 | i7-4790 CPU @ 3.60GHz (8) | 8 + 8 GiB (1600MHz) | 256GB SMART SSD SZ9STE | Gigabit |
| lps12 | i7-4790 CPU @ 3.60GHz (8) | 8 + 8 GiB (1600MHz) | 256GB SMART SSD SZ9STE | Gigabit |

**Table A3.** Cluster software configuration. Updated at shows when each cluster node was last fully updated.

| Node | Operating System (updated at) | Numpy | IPython | pyfftw | Linux Kernel |
|------|-------------------------------|-------|---------|--------|--------------|
| host | Fedora 24 Workstation (17-08-2016) | 1.11.0 | 3.2.1 | 0.10.3.dev0+e827cb5 | 4.6.6-300.fc24.x86_64 |
| lps01 | Fedora 24 Server (16-08-2016) | 1.11.0 | 3.2.1 | 0.10.3.dev0+e827cb5 | 4.6.6-300.fc24.x86_64 |
| lps02 | Fedora 24 Server (16-08-2016) | 1.11.0 | 3.2.1 | 0.10.3.dev0+e827cb5 | 4.6.6-300.fc24.x86_64 |
| lps04 | Fedora 24 Server (16-08-2016) | 1.11.0 | 3.2.1 | 0.10.3.dev0+e827cb5 | 4.6.6-300.fc24.x86_64 |
| lps05 | Fedora 24 Server (16-08-2016) | 1.11.0 | 3.2.1 | 0.10.3.dev0+e827cb5 | 4.6.6-300.fc24.x86_64 |
| lps06 | Fedora 24 Server (16-08-2016) | 1.11.0 | 3.2.1 | 0.10.3.dev0+e827cb5 | 4.6.6-300.fc24.x86_64 |
| lps08 | Fedora 24 Server (16-08-2016) | 1.11.0 | 3.2.1 | 0.10.3.dev0+e827cb5 | 4.6.6-300.fc24.x86_64 |
| lps09 | Fedora 24 Workstation (2016-08-16) | 1.11.0 | 3.2.1 | 0.10.3.dev0+e827cb5 | 4.6.6-300.fc24.x86_64 |
| lps10 | Fedora 24 Server (16-08-2016) | 1.11.0 | 3.2.1 | 0.10.3.dev0+e827cb5 | 4.6.6-300.fc24.x86_64 |
| lps11 | Fedora 24 Server (16-08-2016) | 1.11.0 | 3.2.1 | 0.10.3.dev0+e827cb5 | 4.6.6-300.fc24.x86_64 |
| lps12 | Fedora 24 Server (16-08-2016) | 1.11.0 | 3.2.1 | 0.10.3.dev0+e827cb5 | 4.6.6-300.fc24.x86_64 |

*Appendix A.3. Dataset*

The dataset is composed of 35 neurophysiological signals each with four simultaneously captured channels. The average number of signal samples is about 1 million samples with standard deviation of about 500 thousand samples.

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
