# Peer review of "Parallelism Strategies for Big Data Delayed Transfer Entropy Evaluation"

_algorithms, doi:10.3390/a12090190_

Round 1

Reviewer 1 Report

This paper presented an experiment that explored using parallelism to improve the performance of Delayed Transfer Entropy (DTE) computation. The authors reached conclusion that task parallelism overruns data parallelism in performance boosting.

This information is useful in the literature for those who encounter similar issues with time-consuming computations in data analysis.

There are a few items that the authors should consider:

The speedup of either the task parallel program or the data parallel program itself over the performance of the program without using parallelism should be reported. It only makes sense to compare the performance of the task parallel program and the data parallel program when both of them have speedups over non-parallelized program. The authors used IPython to achieve parallelism. It would be a good idea to compare IPython to Spark and Tensorflow to confirm the reported result.

Reviewer 2 Report

"Parallelism Strategies for Big Data Delayed Transfer Entropy Evaluation"

by Dourado et al. (Id: algorithms-577611)

The authors proposed a novel method using the Delayed Transfer Entropy (DTE) in big data series using heterogeneous Beowulf cluster.

They obtained a significantly faster (Task Parallelism) system in comparison to Data Parallelism. My main discussion on this paper is related to methodology: what can you say about the relationship between TE (or DTE) and Mutual Information? Why you decided to use DTE instead Mutual information?

Also, in order to consider this manuscript to be published in Algorithms journal, please consider the following minor comments/suggestions:

1. L25: "... neuron connectivity and by Arellano-Valle et al. (2013) to optimization of metheorological network design;".
2. L31: Also, TE has been implemented in non-gaussian distributions, such as: multivariate exponential, logistic, Pareto (type I - IV) and Burr distributions (Jafari-Mamaghani and Tyrcha, 2014).
3. L47-48: "After testing DTE feasibility on Beowulf cluster," <-> "Then,".
Put space in: L62 ("processing [43]"), L66 ("hidden [2]"), L255 ("99.9% (n=999)"), and revise all manuscript.
4. L63: "Another definition is proposed by".
5. L105: "presents" <-> "presented".
6. L123: delete "equation" (repeated).
7. L124: delete "presented is".
8. L124-126: Put the lines "where... Figure 1." below of Eq. (1).
9. L127: Section 2.3.2. could moved to materials and methods, because is the novel method to be implemented.
10. L130: "[76] is".
11. L135: "al. [77])," and "analysis (also called".
12: L141: "... X to Y. DTE is defined as".
13: L207: delete "shown in Equation 4.".
12. Put comma at the end of Eq. (3).
13. Put point at the end of Eq. (4) and Fig. (4) caption.

Suggested references

1. Jafari-Mamaghani, M., Tyrcha, J. (2014). Transfer entropy expressions for a class of non-Gaussian distributions. Entropy 16, 1743-1755.
2. Arellano-Valle, R.B., Contreras-Reyes, J.E., Genton, M.G. (2013). Shannon Entropy and Mutual Information for Multivariate Skew-Elliptical Distributions. Scandinavian Journal of Statistics 40, 42-62.
